# Exposure Assessment of Environmental Tobacco Aerosol from Heated Tobacco Products: Nicotine and PM Exposures under Two Limited Conditions

**DOI:** 10.3390/ijerph17228536

**Published:** 2020-11-18

**Authors:** Tomoyasu Hirano, Tokuaki Shobayashi, Teiji Takei, Fumihiko Wakao

**Affiliations:** 1Health Service Division, Health Service Bureau, Ministry of Health, Labour and Welfare, Government of Japan, Tokyo 100-8916, Japan; shobayashi-tokuaki@mhlw.go.jp (T.S.); takei-teiji@mhlw.go.jp (T.T.); 2Center for Cancer Control and Information Services, National Cancer Center, Tokyo 104-0045, Japan; fwakao@ncc.go.jp

**Keywords:** heated tobacco products (HTPs), secondhand aerosol, policy, nicotine, particulate matter 2.5 (PM_2.5_), exposure assessment

## Abstract

It is too early to provide a clear answer on the impact of exposure to the second-hand aerosol of heated tobacco products (HTPs) in the planning of policy for smoke-free indoors legislation. Here, we conducted a preliminary study to evaluate indoor air quality with the use of HTPs. We first measured the concentration of nicotine and particulate matter (PM_2.5_) in the air following 50 puffs in the use of HTPs or cigarettes in a small shower cubicle. We then measured these concentrations in comparison with the use equivalent of smoking 5.4 cigarettes per hour in a 25 m^3^ room, as a typical indoor environment test condition. In the shower cubicle test, nicotine concentrations in indoor air using three types of HTP, namely IQOS, glo, and ploomTECH, were 25.9–257 μg/m^3^. These values all exceed the upper bound of the range of tolerable concentration without health concerns, namely 3 µg/m^3^. In particular, the indoor PM_2.5_ concentration of about 300 to 500 μg/m^3^ using IQOS or glo in the shower cubicle is hazardous. In the 25 m^3^ room test, in contrast, nicotine concentrations in indoor air with the three types of HTP did not exceed 3 μg/m^3^. PM_2.5_ concentrations were below the standard value of 15 μg/m^3^ per year for IQOS and ploomTECH, but were slightly high for glo, with some measurements exceeding 100 μg/m^3^. These results do not negate the inclusion of HTPs within a regulatory framework for indoor tolerable use from exposure to HTP aerosol, unlike cigarette smoke.

## 1. Introduction

Heated tobacco products (HTPs) are tobacco products that produce aerosols containing nicotine and other chemicals which are inhaled by users through the mouth [1]. In order to produce the nicotine-infused vapor, HTPs heat tobacco up to 350 °C (lower than the 700 °C reached with conventional cigarettes) using battery-powered heating systems [2]. Different HTP devices use different heating sources, including electronic energy via battery. The enclosed heating system can be an external heat source to aerosolize nicotine from specially designed cigarettes (e.g., IQOS and glo), or a heated sealed chamber to aerosolize nicotine directly from tobacco leaf (e.g., ploomTECH) [3].

More recently, HTPs have been marketed and promoted by the major tobacco companies as safer alternatives to combustible cigarettes [4]. The use of heated tobacco products (HTPs) has increased rapidly over the last few years in Japan [5]. In 2018, the overall prevalence of monthly HTP use was 2.7% (1.7% daily use) [3]. Virtually all HTP users were current cigarette smokers (67.8%) or former smokers (25.0%), and only 1.0% were never smokers [5]. Many HTPs users (50.5% of HTP users) who switch from conventional cigarettes to HTPs do so to avoid exposing bystanders to second-hand tobacco smoke [6]. Major tobacco companies advertise that HTPs do not generate side-stream smoke or pollute indoor air quality [7]. In contrast, the World Health Organization (WHO) recommends that HTPs should be subject to the same policy and regulatory measures as all other tobacco products, in line with the WHO Framework Convention on Tobacco Control (WHO FCTC) [1,8].

The revised Health Promotion Act is planned for enactment before the Tokyo 2020 Olympic and Paralympic Games to protect against exposure to second-hand tobacco smoke. As stated in Article 8 of the WHO Framework Convention on Tobacco Control (WHO FCTC), legislation for indoor smoking bans relies on sufficient and unequivocal scientific evidence that exposure to tobacco smoke causes death, disease, and disability. Since the evidence available to date does not convincingly demonstrate that the available HTPs will simply replace conventional cigarettes among current smokers without attracting youths, or even that these products will substantially reduce health risks among users [2,9,10], it is regrettably too early to provide a clear answer on the long-term impact of exposure to the second-hand aerosol of heated tobacco products (HTPs) [1,5,7,8]. Scientific evidence has not unequivocally established that exposure to aerosol from HTPs causes death, disease, and disability. So, under the revised act, HTPs are treated differently than cigarettes and are positioned as transitional measure until associations with morbidity or mortality are epidemiologically proven [11]. Some people have argued that HTPs should be regulated in the same way as cigarettes, in accordance with the precautionary principle [12], while others hold that they should remain unregulated unless there is a legal basis for doing so.

Philip Morris International (PMI) studied and reported the impact of IQOS on indoor air quality, evaluated in an environmentally controlled room using ventilation conditions recommended for simulating “office”, “residential”, and “hospitality” environments [13]. As industry results, however, these need to be validated through independent study, using a more appropriate evaluation reflecting actual conditions in Japanese restaurant and bar environments. Further, the results should be compared among three different HTPs sold in Japan, not just IQOS, under the same conditions.

Here, therefore, we conducted a pilot exposure assessment with some risk characterization of nicotine and particulate matter (PM) from HTPs under two different conditions. First, we investigated the legitimacy of health concerns under conditions which aimed to replicate use in a small indoor environment. Second, we measured the impact of using these HTPs on indoor air quality under usual indoor environmental test conditions.

## 2. Methods

### 2.1. No-Observable-Adverse-Effect Level for Nicotine

The no-observable-adverse-effect level (NOAEL) for nicotine was set at 0.5 mg/m^3^, based on a two-year rat inhalation study. A second NOAEL was determined at a dose of 1.25 mg/kg/day; in that 10-day study, rats exhibited mild fatty change, mild focal necrosis, and mild cellular change, with an effect on mitochondria, in a dose-proportional manner. These results were previously reported by Rijksinstituut voor Volksgezondheid en Milieu (The Dutch National Institute for Public Health and the Environment; RIVM) and US Environmental Protection Agency (EPA) [14,15].

### 2.2. Toxicological Risk Assessment of Nicotine

Given a NOAEL of 0.5 mg/m^3^, exposure was converted to that occurring at 24 h per day for 7 days, and to include consideration of those who are generally sensitive to chemical substrates, such as patients with cancer and pulmonary or cardiovascular disease, and small children. This conversion was done by applying an uncertainty factor consisting of a 10-fold interspecies difference and a 10-fold individual difference, which resulted in a calculated tolerable concentration without health concerns of 3.0 μg/m^3^.
0.5 mg/kg/m^3^ × 20/24 × 5/7 × (1/100) = 3.0 μg/m^3^(1)

Additionally, given a daily human respiration rate of 20 m^3^/day and average body weight of 50 kg according to an evaluation guideline [16], uncertainty was high because the NOAEL was a concentration based on a semi-acute toxicity test. Since the NOAEL data obtained from the short-term test period were used, an extra 10-fold uncertainty factor should be introduced. Assuming an overall factor of 1000, the tolerable concentration was calculated to be 3.1 μg/m^3^.
1.25 mg/kg/day × 50 kg ÷ 20 m^3^ × (1/1000) = 3.1 μg/m^3^(2)

### 2.3. Using HTPs and Smoking Cigarette

Three HTPs and one cigarette brand were used. The three HTPs are major brands in Japan and have been characterized in detail elsewhere. For IQOS (Philip Morris Products S.A. Neuchâtel, Switzerland), “Marlboro regular” tobacco sticks called HeatSticks were used. For glo (British American Tobacco plc., London, UK), “Kent” tobacco sticks called neostiks were used. For ploomTECH (Japan Tobacco Inc., Tokyo, Japan), “Mevius regular” liquid capsules called Tobacco caps were used. The devices used were genuine products provided by tobacco companies for each product. A conventional cigarette, Mevius One (Japan Tobacco Inc., Tokyo, Japan), was used as this is the number one selling cigarette product of the number one selling brand in Japan [17].

### 2.4. Concentration of Nicotine

Aerosol phase nicotine in air was captured using ISO 18145:2003, the standard testing procedure [18]. Nicotine was collected in a solid phase, non-polar adsorption resin cartridge sampler XAD-4 (Sigma-Aldrich, St. Louis, MO, USA, Cat No. 54254-U) with a low volume air sampler at the rate of 1.0 L/min. A total of 100 L of air was transferred to the sorbent tubes for both the shower cubicle and room testing. An ozone scrubber (Sigma-Aldrich, St. Louis, MO, USA, Cat No. 505285) was installed inline before the cartridge.

Nicotine was eluted by methanol, and then analyzed and quantified by a liquid chromatography-tandem mass spectrometry (LC/MS/MS) system [19]. The HPLC system consisted of a high-pressure liquid pump, an auto sampler, and a mass spectrometer (AB Sciex QTRAP4500) operated in Multiple Reaction Monitoring (MRM) mode. Ten microliters (10 μL) of sample were injected to purify polar compounds in a hydrophilic interaction liquid chromatography column (KINETEC HILIC φ2.1 mm × 100 mm, particle size 2.6 μm) at 40 °C and a flow rate of 30 mL/min. Nicotine-d_3_ was used as the internal standard for nicotine. Signal output was simultaneously monitored by two ions for nicotine (m/z 163/130), and nicotine-d_3_ (m/z 166/130). The concentration of nicotine in a specimen was determined by calculating the ratio of each ion peak area relative to internal standard peak area. Results from calibrator analysis were used to create a calibration curve using simple linear regression analysis. The slopes and intercepts from the resulting calibration equation were used to calculate control and specimen results. Linearity in the samples was confirmed in the range of 0.0002–0.020 mg/L, i.e., equivalent to 0.025–2.5 μg/m^3^ in air.

### 2.5. Concentration of Particulate Matter 2.5 (PM_2.5_)

Concentrations of PM_2.5_ were measured using Shibata LD-5R (Shibata Scientific Technology Ltd., Soka, Saitama, Japan) and Sidepak TSI AM510 aerosol monitor (TSI Incorporated, Shoreview, MN, USA) in 1-min average mode. PM concentrations were calculated using a mass concentration conversion factor for tobacco smoke (K value) of 0.81 (μg/m^3^/counts per minute) for the Shibata LD-5R [20] and corrected by a conversion factor of 0.295 for the Sidepak TSI AM510 [21]. Averages per minute were measured, and measurement was continued for 120 min after the start of use or smoking in the shower cubicle test and for 60 min in the room test.

### 2.6. Test Shower Cubicle

A shower cubicle (length 0.80 m × width 0.80 m × height 2.24 m) was used as the test room. To prevent ventilation, the ventilation fan and drainage were covered with plastic tape. Measuring instruments were placed on strut-pole shelves set at heights of 1.8 m and 1 m, and then adjusted so that the sample entrance ports of the devices were at the respective heights (Appendix A). The measurement tests were conducted on ploomTECH, glo, IQOS, and conventional cigarette, in that order. The cubicle was well ventilated between test intervals, and every walls, ceiling, floor, and strut-pole shelves were carefully wiped down with a water-soaked rag. The sealing tapes were removed, and the door was left fully open to ventilate the room during intervals. The intervals between each test were about one hour, and next tests were conducted after confirming that the background concentration of PM_2.5_ was confirmed to be below 10 μg/m^3^. Two measurement tests were conducted, including measurements for a preliminary review of the test conditions.

In the shower cubicle test, the subject used or smoked 50 puffs of an HTP or cigarette at 30 s intervals. Testing was done by the same single smoker, a male aged in his 40 s who was an employee of the facility management authority and normally used conventional cigarettes. No other person was present in the shower cubicle during testing. The volume of use and smoking per puff was consistent with the smoker’s normal smoking behavior. The smoker exited the shower cubicle after using or smoking the prescribed number of puffs, in consideration of the adverse effects of exposure, after which the measurements were continued.

### 2.7. Test Room

The Japanese Government previously held an expert review meeting to consider and formulate a standard test environment in 2010 [22]. The report of this meeting specified an average smoking frequency per each smoker of 1.24 cigarettes per hour, determined from the average number for the most frequently consumed smoking product, a population smoking prevalence of 0.218 (21.8%) at the time, and a safety factor of 2, using a room with an effective area of about 25 m^2^, as considered suitable for a restaurant and bar with about 10 seats. Based on these factors, the estimated average number of cigarettes smoked in the room was 5.4 per hour. In the test, basically the same person(s) smoked or used all of the assumed amount.

The study was conducted in a test facility used to evaluate painting-related safety at the Japan Organization of Occupational Health and Safety, in Noborito, Kawasaki City [22]. The room area was 25 m^2^ (length 3.44 m × width 7.26 m). The walls, floor, and ceiling were covered in vinyl sheeting and sealed with tape to prevent air leakage. The height of the room was 2.56 m (Appendix A).

For all measurements, to test the worst-case scenario, the room ventilator was turned off and the test was conducted under non-ventilated conditions. After each test, a powerful horizontal flow push-pull ventilator was operated with a ventilation rate of over 60 times/h and a sufficient reduction in PM_2.5_ concentrations to background level was confirmed before the following test was commenced.

The sorbent tubes in which nicotine was collected and the PM_2.5_ measuring devices were installed 1.2 m from the floor at a distance of 1.5 m and 2.5 m from the smoker, and perpendicular to the direction of the smoker’s exhalation. Three measurement tests were carried out, including measurements for a preliminary review of the test conditions.

In the room test, since nicotine concentrations in mainstream aerosol from the HTP products IQOS, glo, and ploomTECH have already been reported [23,24], we established usage from the number of puffs giving closely similar amounts of nicotine at 54 puffs for IQOS, 130 puffs for glo, 265 puffs for ploomTECH, and 54 puffs for the cigarette, Mevius One (Appendix A). As the mainstream aerosol of IQOS contains nicotine at the same level as cigarette smoke, the number of puffs was set at the same level as that for Mevius One, a low tar cigarette. To account for the large number of puffs required by ploomTECH, testing for this product was done using an additional user (sex: male, age: 40 s, same affiliation), and the two persons each used once every 20 s, in order to finish the use of ploomTECH at roughly the same test time as for the other products.

## 3. Results

### 3.1. Indoor Air Quality in the Small Room

Nicotine and PM_2.5_ concentrations in the shower cubicle test are summarized in Table 1. Among the 3 types of HTP tested, ploomTECH had the lowest nicotine concentration. Concentrations at the heights of 1.0 and 1.8 m were 29.3 and 25.9 μg/m^3^, respectively. Nicotine concentrations using glo and IQOS were one order higher than those with ploomTECH, at 160 and 111 μg/m^3^ for glo and 257 and 212 μg/m^3^ for IQOS, respectively. In contrast, nicotine concentration when smoking a cigarette, Mevius One, was an order of magnitude higher again, at 1040 and 2420 μg/m^3^ at 1.0 and 1.8 m, respectively.

PM_2.5_ also showed the same trend as nicotine for the 3 types of HTP and cigarette, with the lowest values seen for ploomTECH, at 21 and 10 μg/m^3^ (standard deviation (SD) = 55, 6.6) at 1.0 and 1.8 m, respectively, followed by 330 and 99 μg/m^3^ (SD = 564, 119) for glo and 492 and 413 μg/m^3^ (SD = 667, 466) for IQOS, respectively. PM_2.5_ concentration with the cigarette Mevius One exceeded the upper limit of measurement for both instruments at both heights. After exceeding the upper limit of the devices at just over 10 min from the start of the test, levels did not drop below this limit until 120 min after the end of the test.

### 3.2. Indoor Air Quality under the Usual Condition

Nicotine and PM_2.5_ concentrations in the test room are summarized in Table 2. Indoor nicotine concentrations for IQOS and glo at 1.5 and 2.5 m from the user were closely similar, at 2.6 and 2.7 μg/m^3^, and 2.3 and 3.0 μg/m^3^, respectively. Concentrations were lower for ploomTECH, at 0.48 and 0.41 μg/m^3^, respectively. In contrast, concentrations for Mevius One cigarettes were at the high and closely similar levels of 130 and 160 μg/m^3^, and thus not comparable to those with HTPs.

PM_2.5_ tended to differ from nicotine concentration due to the product characteristics of each HTP. PM_2.5_ concentrations at 1.5 and 2.5 m from the user were low for ploomTECH and IQOS, at 6.5 and 7.0 μg/m^3^ (SD = 5.8, 2.7), and 7.0 and 6.9 μg/m^3^ (SD = 11.6, 4.0), respectively, but higher for glo, at 102 and 56 mg/m^3^ (SD = 95, 56), respectively. In contrast, concentrations of PM_2.5_ when Mevius One cigarettes were smoked was 378 and 434 μg/m^3^ (SD = 215, 243), respectively, which were much higher than when HTPs were used.

## 4. Discussion

According to a previously published paper that examined chemical concentrations in mainstream smoke, while some HTPs contain nicotine at similarly high concentrations to cigarettes, levels of carcinogens are generally low [23,24]. Propylene glycol and glycerol are the most common components of PM produced by HTPs [23,24]. Nicotine were detected and assessed from second-hand aerosol in the air, as well as when using electronic cigarettes [25,26]. Nicotine concentrations in the shower cubicle test were at least one order of magnitude greater than the tolerable concentration without health concerns of 3.0 μg/m^3^ with use of any of the three types of HTP, namely ploomTECH, glo, and IQOS, althoughlevels were orders of magnitude lower than those with the cigarette. Many bars and pubs are of very small size and lack adequate ventilation, and patrons and staff often find themselves seated or standing close to each other. The area modeled in our shower cubicle of 0.64 square meters per person appears reasonable. The partially revised Health Promotion Act provides for transitional measures for HTP use under which the long-term health effects of second-hand HTP aerosol have yet to be scientifically proven [11]. Nevertheless, HTPs are included in regulations and their use will be permitted only in designated rooms that meet specified technical requirements. The results of this study are not a legislative basis for the law, but support the inclusion of HTPs within a regulatory framework for indoor use to protect nonsmokers from exposure to HTP aerosol, particularly with regard to the protection of people who are generally sensitive, such as patients with cancer and pulmonary or cardiovascular disease, and small children.

In contrast, nicotine concentrations in the test room remained below 3.0 μg/m^3^ for all HTPs tested, indicating that nicotine is of no concern with the use of these products. Nicotine dependence occurs as three types: physical dependence, habit dependence, and psychological dependence [27]. In this study, assuming physical dependence, normalization was performed against the amount of nicotine in mainstream aerosol/smoke. The inhalation volume of users would be accordingly higher for products with less nicotine in mainstream aerosol/smoke. Nevertheless, for all three HTPs, the nicotine concentration in the room was significantly less than that with cigarettes, and did not exceed the tolerable concentration without health concern level of 3.0 μg/m^3^ at any time. For the other two types of dependence, it is not necessary to normalize against the amount of nicotine, and assuming that these types will be used, nicotine concentration in the room should be further lowered. It should be remembered that levels depend on the strength of ventilation and the structure of the room. Moreover, our testing was conducted with the room ventilator stopped throughout the test, whereas the technical requirement for designated heated tobacco smoking rooms under the law requires ventilation [8]. Nicotine concentration in ventilated rooms would likely decrease further.

Tests conducted by PMI reported nicotine concentrations of 1.10, 1.81, and 0.66 μg/m^3^, respectively, in a room of 24.1 m^2^ in size. Indoor constituents were measured during IQOS sessions under conditions that simulated office, residential, and hospitality environments. When conventional cigarettes were used, namely Marlboro Gold, values were 34.7, 29.1, and 34.6 μg/m^3^, respectively [13]. In our present study, nicotine concentrations did not differ significantly from those in the PMI study of IQOS, although there were some differences in condition settings, such as the presence/absence of ventilation. Specifically, PMI stated “Using THS2.2 (i.e., IQOS) indoors does not have a negative impact on air quality”, “The European Agency for Safety and Health at Work has established a exposure limit at 500 μg/m^3^ (8 h) to be compared to a maximum median value of 1.8 μg/m^3^ for THS 2.2 (275-fold lower)” [28]. For our present study, however, these statements would not necessarily be appropriate. Using HTPs indoor also has negative impacts on air quality. Moreover, while nicotine concentrations in the air are below the tolerable level of 3 μg/m^3^ under controlled scenarios in the normal range, these could be easily exceeded in some conditions, as shown by the shower cubicle test.

The indoor PM2.5 concentration of about 300 to 500 μg/m^3^ when using IQOS or glo in the shower cubicle is in the air quality index (AQI) category of “hazardous” (Figure 1) [29], a level that exceeds the average daily concentration of 350 μg/m^3^ in just 2 h indoors. In contrast, when IQOS and ploomTECH were used in the 25 m^2^ room, levels were below the standard value of 15 μg/m^3^ per year and 35 μg/m^3^ per day, as specified by the US Environmental Protection Agency and Ministry of the Environment, Japan [29,30]. PM_2.5_ concentration when using glo was slightly high, albeit still much lower than the hundreds of micrograms seen when cigarettes are smoked. Some measurements exceeded 100 g/m^3^, but it is unclear whether this would exceed the standard value of 35 μg/m^3^ for 24 h considering the amount of time spent in the room per day. As PM_2.5_ concentration with glo tended to differ from those with IQOS and ploomTECH, research into particulate exposure with HTPs should include the physical properties of fine particles.

Under the revised Health Promotion Act, the handling of HTPs is a transitional measure, namely a provisional measure until a clear answer on the long-term health impact of usage or exposure is obtained [11]. This study did not negate the registration of HTPs under the revised Act. Research into the adverse health effects of HTP use and second-hand aerosol is a public health duty, and the law should be reviewed to take account of future research. Under the revised Health Promotion Act, the regulation of HTPs is a transitional measure. Further revisions of this transitional measure will require additional research on indoor air quality.

Several limitations of our study warrant mention. First, the limited time between the drafting of the bill and its submission to the National Diet, as well as resource constraints, did not allow a sufficient number of tests for statistical analysis of the measurement results. There was not enough time to submit the bill to parliament in order to meet the parliamentary debate schedule. Significant resource constraints were present in terms of testing sites, personnel, and funding, among others. While this is a small-scale test of a hypothesis to obtain a primary outcome, this might raise concerns about the robustness of the methodology. Second, the NOAEL of nicotine was derived from testing in rat models using exposure to inhalation of nicotine [14,15], which might not be extrapolatable to human conditions. Differences in model animal species and the ongoing accumulation of findings from chronic toxicity studies may change assumptions about the tolerable concentration without health concerns. Third, we only investigated two limited cases, and the failure to account for a range of scenarios and variables, including ventilation rate, room conditions and number of smokers, means that our results cannot be generalized. The government study group’s report stipulated only the size of the floor area, as determined from an assumed number of restaurant seats [22]. In our present test, we used a testing room with a height of 2.56 m. Other venues have lower ceiling heights, and concentrations would accordingly be expected to be higher. Actual customer behavior and the manner of smoking of individual smokers or users will vary, requiring evaluation in actual restaurants and bars in future studies. In addition, the room test was conducted under non-ventilated conditions. Measurement under regular ventilation conditions was not possible due to constraints of the test facility. After the legislation on HTPs is enacted, indoor air quality should be measured and evaluated in newly emerging designated HTP smoking rooms in various real-world restaurants, cafés, and other environments. Results on differences in air quality, for example nicotine and PM concentrations, depending on conditions among facilities of different types, room sizes, and ventilation systems could provide insight into how the revised Act’s transitional measures should be reviewed. Fourth, various types and flavors of HTP have been recently put on the market, but this study does not cover them. In addition, counterfeit devices having different operating temperatures from genuine products have also emerged. Evaluation testing of these is an issue for the future. The limitations presented by these test conditions are mitigated by the fact that concerns about second-hand aerosol arise even with HTPs. Finally, and most significantly among limitations, this study did not evaluate other chemicals, such as volatile organic carbons (VOCs), polynuclear aromatic hydrocarbons (PAHs), tobacco-specific nitrosamines (TSNAs), and aldehydes [23,24]. Exposure risk assessment of indoor air environments should assess not only nicotine and particles, but also other toxic chemicals, especially carcinogens [23,24,31,32]. Since existing thresholds do not consider carcinogenicity through injury of genes and mutagenicity etc. through activity on germ cells, one method of risk assessment used in such cases adopts the amount which causes carcinogenesis at a probability of 1/100,000 as a virtually safe dose (VSD) [16,33]. Multiple options are proposed, and this topic remains controversial.

## 5. Conclusions

An exposure assessment of nicotine and particulate matter (PM) from heated tobacco products (HTPs) in two different conditions, namely heavy use and usual environment, was conducted to provide a rough overview of the overall situation with these products. The results of this study are not a legislative basis for the law, but did not negate the inclusion of HTPs within a regulatory framework for indoor use aimed at reducing bystander risk of exposure to HTP aerosol to tolerable levels.

## Figures and Tables

**Figure 1 ijerph-17-08536-f001:**
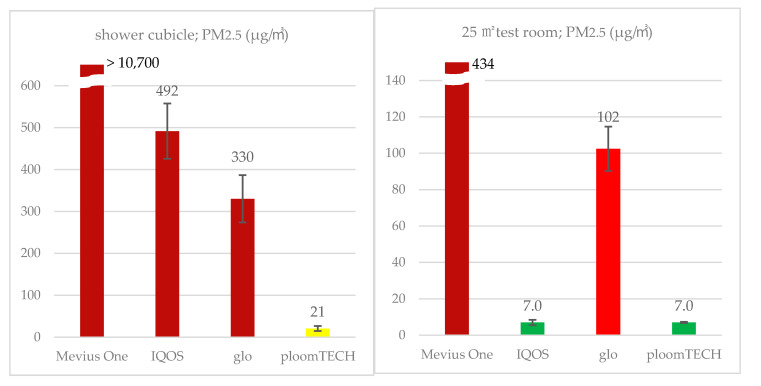
PM_2.5_ concentration in the shower cubicle and the 25 m^2^ testing room. PM_2.5_ concentrations in the shower cubicle and 25 m^2^ testing room are as measured at 1.8 m and 1.0 m in height, whichever was higher; and in the indoor test, whichever was higher at 1.5 m and 2.5 m in distance perpendicular to the direction of exhalation. In the shower cubicle test (left), Mevius One, IQOS, and glo are classified in the AQI category [29] “Hazardous (maroon)” and ploomTECH is in “Moderate (yellow)”. In the 25 m^2^ testing room, in contrast, glo is classified as “Unhealthy (red)”, and IQOS and ploomTECH as “Good (green)”.

**Table 1 ijerph-17-08536-t001:** Nicotine and PM_2.5_ concentration in the shower cubicle test.

HTP	Product	Nicotine (μg/m^3^)		PM_2.5_ (μg/m^3^)			
		1.0 m	1.8 m	1.0 m		1.8 m	
				Mean	SD	Mean	SD
ploomTECH	Mevius regular	29.3	25.9	21	55	10	6.6
Glo	Kent	160	111	330	564	99	119
IQOS	Marlboro regular	257	212	492	667	413	466
(cigarette)	Mevius One	1040	2420	>10,700	-	>5800	-

Nicotine and PM_2.5_ concentration in the shower cubicle (length 0.80 m × width 0.80 m × height 2.24 m) following 50 puffs of HTP or cigarette are shown. Nicotine was collected for 100 min from the time of using/smoking initiation. PM_2.5_ is the mean value of measurements conducted every minute for 120 min. PM_2.5_ concentration for the cigarette Mevius One exceeded the upper measurement limit of the instrument.

**Table 2 ijerph-17-08536-t002:** Nicotine and PM_2.5_ concentrations measured adjacent to the position of the user or smoker.

HTP	Product	Nicotine (μg/m^3^)		PM_2.5_ (μg/m^3^)			
		1.5 m	2.5 m	1.5 m		2.5 m	
				Mean	SD	Mean	SD
ploomTECH	Mevius Regular	0.48	0.41	6.5	5.8	7.0	2.7
Glo	Kent	2.3	3.0	102	95	56	56
IQOS	Marlboro Regular	2.6	2.7	7.0	11.6	6.9	4.0
(cigarette)	Mevius One	130	160	378	215	434	243

Nicotine and PM_2.5_ concentrations in the 25 m^2^ testing room (length 3.44 m × width 7.26 m × height 2.56 m) following smoking of 5.4 cigarettes or equivalent of HTP for 60 min, beginning immediately after the start of use or smoking. Levels were measured 1.5 and 2.5 m from the user, perpendicular to the direction of exhalation.

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
