# Peer review of "Exposure Assessment of Environmental Tobacco Aerosol from Heated Tobacco Products: Nicotine and PM Exposures under Two Limited Conditions"

_ijerph, 2020, doi:10.3390/ijerph17228536_

Round 1
Reviewer 1 Report
An interesting study that analyzes nicotine and PM2.5 concentrations in a room after various “heated tobacco products” and cigarettes are smoked.
I note the following minor concerns:
How often are nicotine and PM2.5 concentrations measured after HTP use?
Please cite reference for the following statement:
LINE 43-44: Virtually all HTP users were current cigarette smokers (67.8%) or former smokers (25.0%); only 1.0% of HTP users were never smokers.
The following statement must be rewritten for clarity.
LINE 59-61: “HTPs were initially established as transitional measures which differed from cigarettes by the National Diet of Japan as a type of tobacco whose associations with morbidity and mortality were not epidemiologically proven”
Author Response
Our responses to the first referee's comments are as follows:
----------------------------------------------------------------------
Response for the First Referee
----------------------------------------------------------------------
How often are nicotine and PM2.5 concentrations measured after HTP use?
For PM2.5, the following are the results. In the shower cubicle test, measurements were taken at one-minute intervals for approximately 10 minutes after 1 hour of ventilation after the test; in the 25 m2 room test, measurements were taken at one-minute intervals for several minutes after the test with strong unidirectional ventilation. This confirmed that the background level had dropped to an order of magnitude µg/m3. For nicotine, background levels were measured after confirming that PM2.5 was dropping.
Please cite reference for the following statement:
LINE 43-44: Virtually all HTP users were current cigarette smokers (67.8%) or former smokers (25.0%); only 1.0% of HTP users were never smokers.
We have cited reference No. 5, as suggested [line 45].
The following statement must be rewritten for clarity.
LINE 59-61: “HTPs were initially established as transitional measures which differed from cigarettes by the National Diet of Japan as a type of tobacco whose associations with morbidity and mortality were not epidemiologically proven”
We have revised the indicated section of text, as suggested [lines 59-62].

Reviewer 2 Report
In this study, the authors conducted a limited study to measure the concentration of nicotine and particulate matter (PM) from using heated tobacco products (HTPs) and smoking cigarettes. The study was conducted both in a small shower cubicle (proxy for heavy use) and in a 25 m3 room (a usual indoor environment). The authors found nicotine concentrations for all HTPs were substantially higher than the tolerable concentration of 3.0 μg/m3 in the shower cubicle test, but below 3.0 μg/m3 in the 25 m3 room test. Indoor PM2.5 concentration for two of the three HTPs (IQOS and glo) in the shower cubicle test was hazardous. The PM2.5 concentration remained high for geo in the 25 m3 room test. These results provide additional evidence for future regulation of indoor HTPs use.
This is a worthwhile topic to investigate, I have a few comments as follows:
- I’d like to see more justification for the premise of the study in the introduction related to the second-hand exposure to HTPs aerosol. Also, why the authors focus on nicotine and PM2.5 rather than other toxic chemicals.
- I have lingering concerns about the representativeness of the two test cases and the external validity of the findings. As the results were based on two limited cases, how would the findings vary with, say, test conditions? I’d like to see additional analysis/discussions on this. This will potentially provide more insights in terms of how future regulations should be best designed and implemented.
Minor points:
- Line 207: “concentrations for IQOS and glo at 1.5 and 2.5 m from the user were closely similar, at 26 and 27” should be “at 2.6 and 2.7”
- Line 213: “but higher for glo, at 105”, not consistent with the number in Table 2 (i.e. 102)
Author Response
Our responses to the second referee's comments are as follows:
----------------------------------------------------------------------
Response for the Second Referee
----------------------------------------------------------------------
Comments:
- I’d like to see more justification for the premise of the study in the introduction related to the second-hand exposure to HTPs aerosol. Also, why the authors focus on nicotine and PM2.5 rather than other toxic chemicals.
According to a previously published paper that examined chemical concentrations in mainstream smoke, while some HTPs contain nicotine at similarly high concentrations to cigarettes, levels of carcinogens are generally low (Bekki K, et al. Journal of UOEH 2017;39, 201-207; Uchiyama S, et al. Chem. Res. Toxicol.2018; 31, 7, 585-593). Propylene glycol and glycerol are the most common components of PM produced by HTPs. By definition, significant amounts of “tar” are produced, but its properties and chemical composition are very different from those of cigarette smoke. We did not think PM would be a more appropriate marker because of its rapid dissipation.
We have added further descriptions to clarify our procedures to readers, as suggested [lines 226-229].
- I have lingering concerns about the representativeness of the two test cases and the external validity of the findings. As the results were based on two limited cases, how would the findings vary with, say, test conditions? I’d like to see additional analysis/discussions on this. This will potentially provide more insights in terms of how future regulations should be best designed and implemented.
We have now described this in detail as a limitation in the Discussion section [lines 321-324].
Minor points:
- Line 207: “concentrations for IQOS and glo at 1.5 and 2.5 m from the user were closely similar, at 26 and 27” should be “at 2.6 and 2.7”
We have corrected the order of values [line 210].
- Line 213: “but higher for glo, at 105”, not consistent with the number in Table 2 (i.e. 102)
We have corrected the number [line 216].

Reviewer 3 Report
The manuscript by Hirano et al reported nicotine and PM2.5 exposure resulting from the use of heated tobacco products (HTPs) in comparison to traditional cigarettes in two testing conditions simulating usual and heavy exposure (relative to the space where they were used). The findings suggested that with a heavy exposure setting, the HTPs can lead to aerosol exposure exceeding safe levels, highlighting the continued need to assess these products for regulatory policymaking. While overall the study methods and findings were well presented, I suggest the authors carefully address some remaining issues so that it can be considered for publication.
Statistical testing of product differences in nicotine and PM2.5 concentrations were not performed. While the authors mentioned this as a limitation, it is not clear why statistical analysis was not performed for paper reporting here. Is it possible that the authors did not have access to original data or due to some other reason?
No-observable-adverse-effect level of nicotine was referred to results based on rat models exposed to inhalational nicotine, which might not be extrapolated to human conditions. The authors also used somewhat arbitrary factors (e.g., 1:100 in Line 87) to account for differences in testing models. The limitations and implications of using such assumptions for data interpretation in this study should be more carefully discussed.
Some sentences are very hard to follow and may be modified for improved readability. For instance, the following sentences may need attention.
In Line 47, “Confounding this” could be removed for better clarity.
Line 54, the sentence “Although…” contains a contrast clause and a main clause, but the contrasting relationship between these clauses is not clear.
Line 140, it is not clear what was meant by “including a condition study”.
Similarly, in Line 169, it is not clear what was referred to by “including a study of conditions”.
In Line 167, the part of sentence “A nicotine…equipment” is hard to understand and should be rewritten.
In Line 247, it is unclear what was meant by “precondition”.
Starting from Line 279, “Research into…indoor air quality.” is repeated in the same paragraph.
In Line 311, “Since… are not considered to have no threshold…casing…”. The double negative is confusing. Casing is a typo of “causing”?
The manuscript should be carefully edited to address other errors or problems, some of which are listed below.
Line 15: limited may be replaced with “preliminary”.
Line 26: a stop is missing before “These results”.
Line 39: e.g should be “e.g.,”
Line 73: tobacco products may be changed to “the HTPs”.
Line 109: missing articles: a high-pressure…, an auto sampler, a mass spectrometer.
Line 116: micotine-d3 should be nicotine-d3.
Line 121: it should be “i.e.,”
Line 146: after leaving should be “after his leaving”
Author Response
Our responses to the third referee's comments are as follows:
----------------------------------------------------------------------
Response for the Third Referee
----------------------------------------------------------------------
Statistical testing of product differences in nicotine and PM2.5 concentrations were not performed. While the authors mentioned this as a limitation, it is not clear why statistical analysis was not performed for paper reporting here. Is it possible that the authors did not have access to original data or due to some other reason?
There was not enough time to submit the bill to parliament in order to meet the parliamentary debate schedule. There were also significant resource constraints in terms of testing sites, personnel, and funds and so on.
We have newly described this in detail as a limitation in the Discussion section [lines 302-304].
No-observable-adverse-effect level of nicotine was referred to results based on rat models exposed to inhalational nicotine, which might not be extrapolated to human conditions. The authors also used somewhat arbitrary factors (e.g., 1:100 in Line 87) to account for differences in testing models. The limitations and implications of using such assumptions for data interpretation in this study should be more carefully discussed.
We have newly described this in detail as a limitation in the Discussion section [lines 306-310].
Some sentences are very hard to follow and may be modified for improved readability. For instance, the following sentences may need attention.
In Line 47, “Confounding this” could be removed for better clarity.
We have revised the indicated section of text, as suggested [line 48].
Line 54, the sentence “Although…” contains a contrast clause and a main clause, but the contrasting relationship between these clauses is not clear.
We have revised the indicated section of text, as suggested [line 55].
Line 140, it is not clear what was meant by “including a condition study”. Similarly, in Line 169, it is not clear what was referred to by “including a study of conditions”.
We have revised the indicated section of text, as suggested [lines 144 and 172-173].
In Line 167, the part of sentence “A nicotine…equipment” is hard to understand and should be rewritten.
We have rewritten the indicated section of text [line 170].
In Line 247, it is unclear what was meant by “precondition”.
We have revised the indicated section of text, as suggested [lines 252-253].
Starting from Line 279, “Research into…indoor air quality.” is repeated in the same paragraph.
We have removed the duplicate part you pointed out [lines 295-299].
In Line 311, “Since… are not considered to have no threshold…casing…”. The double negative is confusing. Casing is a typo of “causing”?
We have revised the indicated section of text, as suggested [lines 332-333], and corrected the spelling [line 334].
The manuscript should be carefully edited to address other errors or problems, some of which are listed below.
Line 15: limited may be replaced with “preliminary”.
We have replaced the term, as suggested [line 16].
Line 26: a stop is missing before “These results”.
We have corrected the noted text [line 27].
Line 39: e.g should be “e.g.,”
We have corrected the noted text [line 40].
Line 73: tobacco products may be changed to “the HTPs”.
We have corrected the noted text [line 77].
Line 109: missing articles: a high-pressure…, an auto sampler, a mass spectrometer.
We have corrected the noted text [lines 114-115].
Line 116: micotine-d3 should be nicotine-d3.
We have corrected the noted text [line 120].
Line 121: it should be “i.e.,”
We have corrected the noted text [line 125].
Line 146: after leaving should be “after his leaving”.
We have revised the indicated section of text with English language editing [lines 149-151].

Round 2
Reviewer 2 Report
I don't have any further comments.